# [1,5]-Hydride Shift Triggered *N*-Dealkylative Cyclization into 2-Oxo-1,2,3,4-tetrahydroquinoline-3-carboxylates via Boronate Complexes

**DOI:** 10.3390/molecules27165270

**Published:** 2022-08-18

**Authors:** Elvira R. Zaitseva, Dmitrii S. Ivanov, Alexander Yu. Smirnov, Andrey A. Mikhaylov, Nadezhda S. Baleeva, Mikhail S. Baranov

**Affiliations:** 1Institute of Bioorganic Chemistry, Russian Academy of Sciences, Miklukho-Maklaya 16/10, 117997 Moscow, Russia; 2Institute of Translational Medicine, Pirogov Russian National Research Medical University, Ostrovitianov 1, 117997 Moscow, Russia

**Keywords:** amides, BF_3_·Et_2_O, debenzylation, nitrogen heterocycles, 2-oxo-1,2,3,4-tetrahydroquinoline-3-carboxylate

## Abstract

A new simple one-pot two-step protocol for the synthesis of 2-oxo-1,2,3,4-tetrahydroquinoline-3-carboxylate from 2-(2-(benzylamino)benzylidene)malonate under the action of BF3·Et2O was developed. It was shown that the reaction proceeds through the formation of a stable iminium intermediate containing a difluoroboryl bridge in the dicarbonyl fragment of the molecule.

## 1. Introduction

The redox economy concept plays an important role in modern organic synthesis and facilitates the efficiency of synthetic pathways [1,2]. The essential tools for such economy are various redox neutral reactions [2]. Incorporation of the reactions into tandem processes is one of the latest trends [3,4,5], which allows increasing molecular complexity in a straightforward and economical manner.

Cyclizations triggered by [1,5]-hydride shift are the most frequently employed variants of the internal redox process [6,7,8,9,10]. This cascade reaction involves the activation of thus an inert C–H bond and results in the formation of various heterocycles (Figure 1, 1st line). For *ortho*-amino benzylidene malonates and other similar derivatives, the transformation proceeds in the presence of various Lewis acids and leads to valuable tetrahydroquinolines (Figure 1) [11,12,13]. Tandem processes exploiting such a reaction are mostly limited to double 1,5-hydride shift triggered cyclization [14,15,16,17]. However, a few reactions involving more complex multistep transformations were recently reported (Figure 1, 2nd and 3rd lines) [18,19]. In the first report, the hydride shift process can be also accompanied by the cleavage of a C–N bond and recyclization into the internal amide via an attack on the carbonyl group (Figure 1) [18]. In the second report, the same recyclization takes place, but the amino substituent still remains in the molecule [19]. These transformations were reported to proceed only with strong acceptor functions, such as Meldrum’s acid or 1,3-dicarbonyl derivatives.

In our previous work [20], we showed that BF_3_·Et_2_O can induce a hydride shift and subsequent cyclization of benzylidene-malonates containing a thioether group, which was shown to be a very poor hydride donor (Figure 1, 4th line). The key factor driving this process is the formation of chelate species with the O–BF_2_–O bridge, which compensates problems in the formation of an intermediate thionium cation (Figure 1, 4th line). The reaction of this Lewis acid with similar amino derivatives with *N*-benzyl fragment leads to a more rapid consumption of the initial malonate [20]. In the present work, we studied this process in more detail and showed that the reaction with BF_3_·Et_2_O results in the formation of a stable iminium cation containing a difluoroboryl bridge. Treatment of this product with water leads to hydrolysis with formal *N*-dealkylation and formation of 2-oxotetrahydroquinoline-3-carboxylate or its boronate complex. It should be noted that no similar transformation of the dialkyl malonate derivatives was feasible in earlier reports [6,8,9], which demonstrates the efficiency of the activation approach via the boronate complex. The developed protocol allows the redox- and step-economical synthesis of 2-oxo-1,2,3,4-tetrahydroquinoline-3-carboxylates.

## 2. Results and Discussion

In a previous work [20], using quantum mechanical calculations, we showed that the [1,5]-hydride shift reaction of benzylidene malonates with boron trifluoride requires two BF_3_ molecules and proceeds via the formation of a stable cation and a difluoroboryl bridge (Figure 1). Such species were especially stable in case of an iminium cation and are probably capable to undergo various other reactions. This prompted us to carry out more detailed studies.

We showed that the prolonged action of the excess of boron trifluoride on malonate derivative **1a**, followed by aqueous treatment, leads to the formation of product **2a*** (Figure 2). In addition, a noticeable amount of benzaldehyde and product **2a** was observed in the mixture. The amount of **2a** increased with prolonged treatment with water or upon purification on silica gel (Figure 2).

The results of the ^1^H NMR spectroscopic analysis of the reaction mixture (Appendix A) revealed the formation of the iminium cationic species **II** with N^+^=CHPh fragment. The presence of such a fragment was confirmed by the appearance of a signet signal at 9.4 ppm, which significantly differs from the signal of the initial **1a** (8.0 ppm for CH=C(CO_2_Me)_2_ in dichloroethane without additional reference). Formation of a difluoroboryl bridge and BF_4_^−^ anion was confirmed by heteronuclear NMR (Appendix A). Similarly to the previously reported data [20,21], the transformation of BF_3_·Et_2_O into the O–BF_2_–O bridge and BF_4_^−^ anion results in the appearance of novel signals: −150.3, −148.4, and −142.4 ppm in ^19^F as well as +0.5 and −1.1 ppm in ^11^B spectra, contrary to −152.7 and −0.2 ppm signals of the initial BF_3_·Et_2_O.

The subsequent aqueous treatment leads to hydrolysis yielding benzaldehyde and secondary amino derivative **IV**, which readily converts into tetrahydroquinoline **2a*** (Figure 3).

Under these conditions, substituted malonates **1** can be easily converted into product **2***. However, most of them were even less stable than **2a***. Therefore, we added a HCl treatment step, which provided pure deboronated product **2**. In addition, we briefly examined the conditions of the first step and found that the best results could be obtained for various malonates **1** when 2.5 molar excess of BF_3_ was used and the reaction proceeded for 24 h upon heating at 60 °C. We found that the formation of such a product was not observed if various metal triflates or other Lewis acids (AlCl_3_, TiCl_4_ or SnCl_4_) were used. The action of triflates results in “classical” 1,5-hydride shift triggered cyclization, while stronger action of Lewis acids results in the formation of more complex mixtures. However, the presence of product **2** was also not observed. We also examined the conditions for [1,5]-hydride shift triggered N-dealkylative cyclization (reported for Meldrum’s acid derivatives) by heating with morpholine in ethanol [18]. Heating of neither derivative **1a** nor a mixture of the corresponding aldehyde with diethylmalonate with morpholine did not lead to the formation of tetrahydraquinoline derivatives—the starting materials remained intact even after 24 h (Figure 2, bottom).

Next, using the revealed conditions, we obtained a series of compound **2** (Figure 3).

In contrast to the previously studied reaction of the sulfur derivatives [20], we did not observe any difference between the derivatives of dimethyl- (**1a**) and diethylmanolanate (**1b**). Aromatic substituents can influence the stability of boronate complex **2*,** but the HCl workup step provided the desired product **2** in good yield in most cases. The exceptions were 8-chloro- (**2d**) and 7-trifluoromethyl- (**2j**) derivatives, which were obtained in the decreased yields of 41% and 45%, respectively. N-benzyl-N-ethyl- substrates **1l** and **1m** provided the target products **2l** and **2m**, respectively, containing an ethyl substituent at the nitrogen atom. In comparison with Mori’s report [16], the recombination process was reasonably efficient also for diethylamino derivative **1n** (giving the same product as ethylbenzylamine derivative **1o**). This means that the described method was not limited to “debenzylation”.

All the revealed limitations of the reaction correlate well with the stability of the iminium cation. In particular, compounds **1p** and **1r** upon a treatment with BF_3_·Et_2_O underwent [1,5]-hydride shift and subsequent cyclization to give compounds **3p** and **3r** in good yield. In both cases, the resulting iminium cations were rather unstable, which shifted the equilibrium toward the classical cyclization product.

## 3. Materials and Methods

### 3.1. Materials

Commercially available reagents were used without additional purification. E. Merck Kieselgel 60 (Merck, Darmstadt, Germany) was used for column chromatography. Thin-layer chromatography (TLC) was performed on silica gel 60 F_254_ glass-backed plates (Merck, Darmstadt, Germany). Visualization was performed using UV light (254 or 312 nm) or by staining with KMnO_4_.

NMR spectra were recorded on a 700 MHz Bruker Avance III NMR (Bruker, Rheinstetten, Germany) at 303K, Bruker Avance III 800 (Bruker, Rheinstetten, Germany) (with a 5 mm CPTXI cryoprobe), and Bruker Fourier 300(Bruker, Rheinstetten, Germany). Chemical shifts were reported relative to the residue peaks of DMSO-*d*_6_ (2.51 ppm for ^1^H and 39.5 ppm for ^13^C). Melting points were measured on an SMP 30 (Buch & Holm A/S, Herlev, Denmark) apparatus without correction. High-resolution mass spectra (HRMS) were recorded on AB Sciex TripleTOF^®^ 5600+ System (AB Sciex, Framingham, MA, USA) using electrospray ionization (ESI). The measurements were performed in a positive ion mode (interface capillary voltage 5500 V); the mass ranged from *m*/*z* 50 to *m*/*z* 3000; external or internal calibration was performed with an ESI Tuning Mix, (Agilent, Santa-Clara, CA, USA). A syringe injection was used for solutions in acetonitrile, methanol, or water (flow rate of 20 μL/min). Nitrogen was applied as a dry gas; the interface temperature was set at 180 °C. IUPAC compound names were generated using ChemDraw Software (PerkinElmer, Waltham, MA, USA).

### 3.2. Experimental Procedures

#### 3.2.1. Synthesis of *Methyl2-((difluoroboranyl)oxy)-1-methyl-1,4-dihydroquinoline-3-carboxylate* (**2a***)

Compound **1a** (1 mmol) was dissolved in dry C_2_H_4_Cl_2_ (5 mL) under argon atmosphere. Freshly distilled BF_3_·Et_2_O (355 mg, 2.5 mmol) was added dropwise, and the resulting mixture was stirred at 25 °C for 24 h. An aqueous solution of NaHCO_3_ (3%, 50 mL) was added, and the resulting mixture was extracted with EtOAc (3 × 50 mL). Combined organic layers were washed with brine (3 × 50 mL), dried over anhydrous Na_2_SO_4_. All volatiles were removed in vacuo, and the residue was purified by flash chromatography (an eluent mixture of hexane and EtOAc, v/v 10:1). Yield 144 mg (54%), white solid, m.p. 188–190 °C.

^1^H NMR (700 MHz, DMSO-d_6_) δ ppm: 3.39 (s, 3 H), 3.70 (s, 2 H), 3.97 (s, 3 H), 7.15 (*td*, *J* = 7.4, 0.9 Hz, 1 H), 7.23 (d, *J* = 8.0 Hz, 1 H), 7.26 (*dd*, *J* = 7.5, 1.0 Hz, 1 H), and 7.28–7.31 (m, 1 H); ^13^C NMR (176 MHz, DMSO-*d*_6_) δ ppm: 22.7, 30.1, 55.2, 72.3, 115.8, 122.4, 124.9, 127.5, 128.9, 136.8, 163.7, and 168.7. HRMS (ESI-TOF) found, *m*/*z*: 268.0957 [M+H]^+^. C_12_H_13_BF_2_NO^3+^. Calculated, *m*/*z*: 268.0951.

#### 3.2.2. General Procedure for Synthesis of the Compounds **2**

The corresponding substance **1** (1 mmol) was dissolved in dry C_2_H_4_Cl_2_ (5 mL) under argon atmosphere. Freshly distilled BF_3_·Et_2_O (355 mg, 2.5 mmol) was added dropwise, and the resulting mixture was stirred at 60 °C for 24 h and cooled to 25 °C. A mixture of 5.4 M solution of HCl in dioxane (0.46 mL, 2.5 mmol) and MeOH (5 mL) was added dropwise, and the resulting mixture was stirred for 6 h at 25 °C. An aqueous solution of NaHCO_3_ (3%, 50 mL) was added, and the resulting mixture was extracted with EtOAc (3 × 50 mL). The combined organic layers were washed with brine (3 × 50 mL), dried over anhydrous Na_2_SO_4_. All volatiles were removed in vacuo, and the residue was purified by column chromatography (an eluent mixture of hexane and EtOAc, *v*/*v* 5:1).

*Methyl 1-methyl-2-oxo-1,2,3,4-tetrahydroquinoline-3-carboxylate* (**2a**). Yield 173 mg (79%), light green solid, m.p. 83–85 °C; ^1^H NMR (700 MHz, DMSO-*d*_6_) δ ppm: 3.09–3.18 (m, 2H), 3.28 (s, 3H), 3.62 (s, 3H), 3.72 (*dd*, *J*=9.4, 6.4 Hz, 1H), 7.03 (*td*, *J* = 7.4, 0.8 Hz, 1H), 7.12 (*d*, *J* = 8.0 Hz, 1H), 7.24 (*d*, J = 7.3 Hz, 1H), and 7.27–7.30 (m, 1H); ^13^C NMR (75 MHz, DMSO-d_6_) δ ppm: 27.9, 29.5, 47.3, 52.1, 115.1, 122.9, 124.0, 127.7, 127.9, 139.6, 165.8, and 169.8; HRMS (ESI-TOF) found, *m*/*z*: 220.0969 [M+H]+. C_12_H_14_NO_3_^+^. Calculated, *m*/*z*: 220.0968. This corresponds to literature data [22].

*Ethyl 1-methyl-2-oxo-1,2,3,4-tetrahydroquinoline-3-carboxylate* (**2b**). Yield 186 mg (80%), pale viscous oil; ^1^H NMR (700 MHz, DMSO-d_6_) δ ppm: 1.12 (t, *J* = 7.1 Hz, 3H), 3.09–3.16 (m, 2H), 3.28 (s, 3H), 3.67 (*dd*, *J* = 8.2, 7.3 Hz, 1H), 4.02–4.12 (m, 2H), 7.03 (t, *J* = 7.3 Hz, 1H), 7.12 (*d*, *J* = 8.0 Hz, 1H), 7.24 (*d*, *J* = 7.3 Hz, 1H), and 7.28 (t, *J* = 7.8 Hz, 1H); ^13^C NMR (75 MHz, DMSO-d_6_) δ ppm: 14.0, 28.0, 29.5, 47.4, 60.7, 115.0, 122.8, 124.0, 127.7, 127.9, 139.6, 165.9, and 169.3; HRMS (ESI-TOF) found, *m*/*z*: 234.1125 [M+H]^+^. C_13_H_16_NO_3_^+^. Calculated, *m*/*z*: 234.1125.

*Methyl 1,6-dimethyl-2-oxo-1,2,3,4-tetrahydroquinoline-3-carboxylate* (**2c**). Yield 158 mg (68%), pink solid, m.p. 108–110 °C; ^1^H NMR (700 MHz, DMSO-*d*_6_) δ ppm: 2.25 (s, 3H), 3.03–3.13 (m, 2H), 3.25 (s, 3H), 3.62 (s, 3H), 3.68 (*dd*, *J* = 9.5, 6.3 Hz, 1H), 7.01 (*d*, *J* = 8.2 Hz, 1H), 7.05 (s, 1H), and 7.08 (*d*, *J* = 8.2 Hz, 1H); ^13^C NMR (75 MHz, DMSO-*d*_6_) δ ppm: 20.2, 27.9, 29.5, 47.3, 52.2, 115.0, 123.8, 128.0, 128.4, 131.9, 137.2, 165.6, and 169.8; HRMS (ESI-TOF) found, *m*/*z*: 234.1126 [M+H]^+^. C_13_H_16_NO_3_^+^. Calculated, *m*/*z*: 234.1125.

*Methyl 8-chloro-1-methyl-2-oxo-1,2,3,4-tetrahydroquinoline-3-carboxylate* (**2d**). Yield 103 mg (41%), white solid, m.p. 107–109 °C; ^1^H NMR (700 MHz, DMSO-*d*_6_) δ ppm: 3.08–3.18 (m, 2H), 3.34 (s, 3H), 3.61 (s, 3H), 3.74 (*dd*, *J* = 9.7, 5.3 Hz, 1H), 7.11 (t, *J* = 7.8 Hz, 1H), 7.27 (*d*, *J* = 7.4 Hz, 1H), and 7.37 (*d*, *J* = 8.0 Hz, 1H); ^13^C NMR (75 MHz, DMSO-*d*_6_) δ ppm: 28.5, 35.9, 47.7, 52.2, 122.8, 125.5, 126.7, 130.0, 131.0, 138.3, 168.2, and 169.0; HRMS (ESI-TOF) found, *m*/*z*: 254.0579 [M+H]^+^. C_12_H_13_ClNO_3_^+^. Calculated, *m*/*z*: 254.0578.

*Methyl 7-chloro-1-methyl-2-oxo-1,2,3,4-tetrahydroquinoline-3-carboxylate* (**2e**). Yield 204 mg (81%), light green solid, m.p. 93–95 °C; ^1^H NMR (700 MHz, DMSO-*d*_6_) δ ppm: 3.12 (s, 1H), 3.13 (s, 1H), 3.27 (s, 3H), 3.63 (s, 3H), 3.75 (t, *J* = 7.8 Hz, 1H), 7.09 (*dd*, *J* = 7.9, 2.0 Hz, 1H), 7.19 (*d*, *J* = 1.9 Hz, 1H), and 7.27 (*d*, *J* = 8.0 Hz, 1H); ^13^C NMR (75 MHz, DMSO-*d*_6_) δ ppm: 27.3, 29.6, 47.0, 52.3, 115.1, 122.4, 123.0, 129.3, 132.1, 141.0, 165.7, and 169.5; HRMS (ESI-TOF) found, *m*/*z*: 254.0583 [M+H]^+^. C_12_H_13_ClNO_3_^+^. Calculated, *m*/*z*: 254.0578.

*Methyl 6-chloro-1-methyl-2-oxo-1,2,3,4-tetrahydroquinoline-3-carboxylate* (**2f**). Yield 173 mg (68%), white solid, m.p. 139–141 °C. ^1^H NMR (700 MHz, DMSO-*d*_6_) δ ppm: 3.10–3.18 (m, 2H), 3.26 (s, 3H), 3.63 (s, 3H), 3.75 (*dd*, *J* = 9.1, 6.8 Hz, 1H), 7.14 (*d*, *J* = 8.8 Hz, 1H), 7.33 (*dd*, *J*=8.6, 2.5 Hz, 1H), and 7.35 (*d*, *J* = 2.5 Hz, 1H); ^13^C NMR (75 MHz, DMSO-*d*_6_) δ ppm: 27.5, 29.6, 46.8, 52.3, 116.8, 126.4, 126.8, 127.3, 127.5, 138.6, 165.6, and 169.5; HRMS (ESI-TOF) found, *m*/*z*: 254.0582 [M+H]^+^. C_12_H_13_ClNO_3_^+^. Calculated, *m*/*z*: 254.0578.

*Methyl 7-bromo-1-methyl-2-oxo-1,2,3,4-tetrahydroquinoline-3-carboxylate* (**2g**). Yield 234 mg (79%), white solid, m.p. 127–129 °C; ^1^H NMR (700 MHz, DMSO-*d*_6_) δ ppm: 3.10 (s, 1H), 3.11 (s, 1H), 3.27 (s, 3H), 3.63 (s, 3H), 3.75 (t, *J* = 7.7 Hz, 1H), 7.19–7.24 (m, 2H), and 7.31 (s, 1H); ^13^C NMR (75 MHz, DMSO-*d*_6_) δ ppm: 27.4, 29.6, 46.9, 52.2, 117.8, 120.3, 123.4, 125.4, 129.6, 141.2, 165.7, and 169.5; HRMS (ESI-TOF) found, *m*/*z*: 298.0071 [M+H]^+^. C_12_H_13_BrNO_3_^+^. Calculated, *m*/*z*: 298.0073.

*Methyl 6-bromo-1-methyl-2-oxo-1,2,3,4-tetrahydroquinoline-3-carboxylate* (**2h**). Yield 213 mg (72%), white solid, m.p. 141–143 °C; ^1^H NMR (700 MHz, DMSO-*d*_6_) δ ppm: 3.11–3.18 (m, 2H), 3.26 (s, 3H), 3.64 (s, 3H), 3.75 (dd, *J*=9.2, 6.7 Hz, 1H), 7.08 (*d*, *J* = 8.6 Hz, 1H), 7.45 (*dd*, *J* = 8.6, 2.3 Hz, 1H), and 7.47 (*d*, *J* = 1.9 Hz, 1H); ^13^C NMR (75 MHz, DMSO-*d*_6_) δ ppm: 27.4, 29.6, 46.8, 52.3, 114.8, 117.2, 126.7, 130.2, 130.3, 139.0, 165.6, and 169.5; HRMS (ESI-TOF) found, *m*/*z*: 298.0075 [M+H]^+^. C_12_H_13_BrNO_3_^+^. Calculated, *m*/*z*: 298.0073.

*Methyl 7-methoxy-1-methyl-2-oxo-1,2,3,4-tetrahydroquinoline-3-carboxylate* (**2i**). Yield 179 mg (72%), colorless viscous oil; ^1^H NMR (700 MHz, DMSO-*d*_6_) δ ppm: 3.01–3.09 (m, 2H), 3.27 (s, 3H), 3.62 (s, 3H), 3.67 (*dd*, *J* = 9.2, 6.5 Hz, 1H), 3.76 (s, 3H), 6.61 (*dd*, *J* = 8.2, 2.3 Hz, 1H), 6.67 (*d*, *J* = 2.3 Hz, 1H), and 7.14 (*d*, *J* = 8.2 Hz, 1H); ^13^C NMR (75 MHz, DMSO-*d*_6_) δ ppm: 27.2, 29.6, 47.6, 52.1, 55.3, 102.2, 107.4, 115.9, 128.5, 140.6, 159.0, 165.9, and 169.8; HRMS (ESI-TOF) found, *m*/*z*: 250.1076 [M+H]^+^. C_13_H_16_NO_4_^+^. Calculated, *m*/*z*: 250.1074.

*Methyl 1-methyl-2-oxo-7-(trifluoromethyl)-1,2,3,4-tetrahydroquinoline-3-carboxylate* (**2j**). Yield 129 mg (45%), colorless viscous oil; ^1^H NMR (700 MHz, DMSO-*d*_6_) δ ppm: 3.23 (*br.d*., *J* = 7.8 Hz, 2H), 3.33 (s, 3H), 3.64 (s, 3H), 3.81 (t, *J* = 7.9 Hz, 1H), 7.38 (s, 1H), 7.40 (*d*, *J* = 7.8 Hz, 1H), and 7.48 (*d*, *J* = 7.8 Hz, 1H); ^13^C NMR (201 MHz, DMSO-*d*_6_) δ ppm: 27.7, 29.6, 46.6, 52.2, 111.4 (q, *J*=3.7 Hz), 119.4 (q, *J* = 3.7 Hz), 124.1 (q, *J* = 272.2 Hz), 128.3 (q, *J* = 32.3 Hz), 128.6, 128.7, 128.8, 140.4, 165.6, and 169.4; HRMS (ESI-TOF) found, *m*/*z*: 288.0844 [M+H]^+^. C_13_H_13_F_3_NO_3_^+^. Calculated, *m*/*z*: 288.0842.

*Methyl 1-methyl-2-oxo-6-(trifluoromethyl)-1,2,3,4-tetrahydroquinoline-3-carboxylate* (**2k**). Yield 172 mg (60%), colorless viscous oil; ^1^H NMR (700 MHz, DMSO-*d*_6_) δ ppm: 3.24 (d, *J* = 7.8 Hz, 2H), 3.32 (s, 3H), 3.64 (s, 3H), 3.83 (t, *J* = 8.0 Hz, 1H), 7.31 (*d*, *J* = 8.4 Hz, 1H), and 7.61–7.67 (m, 2H); ^13^C NMR (201 MHz, DMSO-*d*_6_) δ ppm: 27.4, 29.7, 46.7, 52.2, 115.4, 123.0 (q, *J* = 30.8 Hz), 124.3 (q, *J* = 271.4 Hz), 124.6 (q, *J* = 4.4 Hz), 124.8 (q, *J* = 4.4 Hz), 125.1, 142.9, 165.9, and 169.4; HRMS (ESI-TOF) found, *m*/*z*: 288.0846 [M+H]^+^. C_13_H_13_F_3_NO_3_^+^. Calculated, *m*/*z*: 288.0842.

*Methyl 6-bromo-1-ethyl-2-oxo-1,2,3,4-tetrahydroquinoline-3-carboxylate* (**2l**). Yield 190 mg (61%), white solid, m.p. 96–98 °C; ^1^H NMR (700 MHz, DMSO-*d*_6_) δ ppm: 1.11 (t, *J* = 7.1 Hz, 3H), 3.09–3.16 (m, 2H), 3.63 (s, 3H), 3.74 (*dd*, *J* = 9.1, 6.6 Hz, 1H), 3.90 (q, *J* = 7.3 Hz, 2H), 7.12 (*d*, *J* = 8.8 Hz, 1H), 7.44 (*dd*, *J* = 8.8, 2.3 Hz, 1H), and 7.48 (*d*, *J* = 2.1 Hz, 1H); ^13^C NMR (75 MHz, DMSO-*d*_6_) δ ppm: 12.3, 27.5, 37.1, 46.8, 52.2, 114.6, 117.0, 127.1, 130.3, 130.7, 137.8, 165.1, and 169.5; HRMS (ESI-TOF) found, *m*/*z*: 312.0235 [M+H]^+^. C_13_H_15_BrNO_3_^+^. Calculated, *m*/*z*: 312.0230.

*Methyl 1-ethyl-7-methoxy-2-oxo-1,2,3,4-tetrahydroquinoline-3-carboxylate* (**2m**). Yield 142 mg (54%), colorless viscous oil; ^1^H NMR (700 MHz, DMSO-*d*_6_) δ ppm: 1.12 (t, *J* = 7.1 Hz, 3H), 2.99–3.07 (m, 2H), 3.61 (s, 3H), 3.66 (*dd*, *J* = 9.0, 6.5 Hz, 1H), 3.76 (s, 3H), 3.91 (q, *J* = 7.1 Hz, 2H), 6.61 (*dd*, *J* = 8.2, 2.5 Hz, 1H), 6.67 (*d*, *J* = 2.3 Hz, 1H), and 7.14 (*d*, *J* = 8.2 Hz, 1H); ^13^C NMR (75 MHz, DMSO-*d*_6_) δ ppm: 12.4, 27.3, 37.0, 47.6, 52.1, 55.3, 102.0, 107.2, 116.2, 128.9, 139.3, 159.1, 165.5, and 169.8; HRMS (ESI-TOF) found, *m*/*z*: 264.1238 [M+H]^+^. C_14_H_18_NO_4_^+^. Calculated, *m*/*z*: 264.1230.

*Methyl 1-ethyl-2-oxo-1,2,3,4-tetrahydroquinoline-3-carboxylate* (**2n**). Yield 109 mg (47%) from **1n** and 132 mg (57%) from **1o**, pale viscous oil; ^1^H NMR (700 MHz, DMSO-*d*_6_) δ ppm: 1.13 (t, *J* = 7.1 Hz, 3H), 3.06–3.15 (m, 2H), 3.62 (s, 3H), 3.70 (*dd*, *J* = 9.4, 6.4 Hz, 1H), 3.87–3.96 (m, 2H), 7.00–7.04 (m, 1H), 7.16 (*d*, *J* = 8.0 Hz, 1H), 7.24 (*d*, *J* = 7.4 Hz, 1H), and 7.26–7.29 (m, 1H); ^13^C NMR (75 MHz, DMSO-*d*_6_) δ ppm: 12.5, 28.0, 37.0, 47.3, 52.1, 114.9, 122.8, 124.3, 127.8, 128.2, 138.4, 165.3, and 169.7; HRMS (ESI-TOF) found, *m*/*z*: 234.1130 [M+H]+. C_13_H_16_NO_3_^+^. Calculated, *m*/*z*: 234.1125.

*Dimethyl 1-methyl-6-nitro-2-phenyl-1,4-dihydroquinoline-3,3(2H)-dicarboxylate* (**3p**). Yield 310 mg (81%), yellow solid, m.p. 187–189 °C. ^1^H NMR (700 MHz, DMSO-*d*_6_) δ ppm: 3.01 (s, 3H), 3.12 (*d*, *J* = 16.4 Hz, 1H), 3.28 (m,1H), 3.59 (*d*, *J* = 6.3 Hz, 6H), 5.22 (*d*, *J* = 1.5 Hz, 1H), 6.79 (*d*, *J* = 9.4 Hz, 1H), 7.02 (*dd*, *J* = 7.3, 1.9 Hz, 2H), and 7.32–7.36 (m, 3H), 8.02–8.06 (m, 2H); ^13^C NMR (75 MHz, DMSO-*d*_6_) δ ppm: 27.8, 38.2, 53.0, 53.3, 55.6, 65.1, 109.2, 116.6, 125.1, 125.3, 127.3, 128.7, 128.7, 135.9, 137.6, 149.3, 167.5, and 168.8; HRMS (ESI-TOF) found, *m*/*z*: 385.1393 [M+H]^+^. C_20_H_21_N_2_O_6_^+^. Calculated, *m*/*z*: 385.1394.

*Dimethyl 1-methyl-1,4-dihydroquinoline-3,3(2H)-dicarboxylate* (**3r**). Yield 166 mg (63%), yellow solid, m.p. 97–100 °C; ^1^H NMR (700 MHz, CDCl_3_) δ ppm: 2.91 (s, 3H), 3.30 (s, 2H), 3.62 (s, 2H), 3.74 (s, 6H), 6.60 (*d*, *J* = 8.2 Hz, 1H), 6.69 (t, *J* = 7.2 Hz, 1H), 7.04 (*d*, *J* = 7.4 Hz, 1H), and 7.10 (t, *J* = 7.7 Hz, 1H); ^13^C NMR (75 MHz, CDCl_3_) δ ppm: 33.4, 39.0, 52.8, 52.9, 54.5, 111.2, 117.3, 119.7, 127.3, 128.9, 145.1, and 170.2; HRMS (ESI-TOF) found, *m*/*z*: 264.1235 [M+H]+. C_14_H_18_NO_4_^+^. Calculated, *m*/*z*: 264.1230.

## 4. Conclusions

We developed a new, redox-neutral method for the synthesis of 2-oxo-1,2,3,4-tetrahydroquinoline-3-carboxylates from 2-(*N*-benzyl-*N*-alkylamino)benzylidene malonates. The process was based on the activation of a substrate with two equivalents of boron trifluoride. This activation leads to [1,5]-hydride shift and the formation of a stable iminium intermediate containing a difluoroboryl bridge. The formation of such an O–BF_2_–O bridge was confirmed by a heteronuclear NMR study, while the presence of a stable iminium cation was confirmed by the ^1^H NMR analysis of a reaction mixture. This product undergoes cyclization upon hydrolysis, resulting in the formation of an amide product. In sum, this process can be described as [1,5]-hydride shift triggered N-dealkylative cyclization. The revealed transformation differs from the previously presented examples of hydride shift triggered *N*-dealkylative cyclization, as it does not require the presence of strong electron-accepting functions [18,19], and it is not accompanied with decarboxylation [18] or substituent rearrangements [19].

## Data Availability

All data are contained within the article or Appendix A.

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
