# Peer review of "[1,5]-Hydride Shift Triggered *N*-Dealkylative Cyclization into 2-Oxo-1,2,3,4-tetrahydroquinoline-3-carboxylates via Boronate Complexes"

_molecules, 2022, doi:10.3390/molecules27165270_

Round 1
Reviewer 1 Report
The paper is properly accepted, however I want to know the selectivity arises to use other catalysts rather than triflates.
The conclusion should be improved (require more declare)
Some typist mistakes:
1- DMSO-6 should be DMSO-d6 and d6 should be in italic font
2- m/z should be in italic font
3- In general the references number should be put before "." or "," not after
4- ortho should be in italic fonts
5- we have shown that in line 37 should be we showed......
6- The key factor driving this 39 process to proceed is the should be "The key factor driving this 39 process to proceed, is the.......
7- dimethyl and diethylmanolane should be "dimethyl- and diethylmanolane derivatives....."
8- In our previous work,[20] should be In our previous work [20],
9- 3% aqueous solution of NaHCO3 should be "Aqueous solution of NaHCO3 (3%, 50 mL)......"
I think the paper can be accepted after answering the assigned questions and revising the paper very seriously
Author Response
The paper is properly accepted, however I want to know the selectivity arises to use other catalysts rather than triflates.
>>Only boron trifluoride is able to promote such a transformation. Metal triflates result in clean “classical” 1,5-hydride shift triggered cyclization. AlCl3, TiCl4 or SnCl4 results in formation of more complex mixtures, but “classical” product remains dominant and no formation of compounds 2 were observed. Corresponding discussion was included in the manuscript.
The conclusion should be improved (require more declare)
>> Conclusion section was rewritten. More details and declarations were included.
Some typist mistakes:
- DMSO-6 should be DMSO-d6 and d6 should be in italic font
>> Corrected
- m/z should be in italic font
>> Corrected
- In general the references number should be put before "." or "," not after
- ortho should be in italic fonts
>> Corrected
- we have shown that in line 37 should be we showed......
>> Corrected
- The key factor driving this 39 process to proceed is the should be "The key factor driving this 39 process to proceed, is the.......
>> Corrected
- dimethyl and diethylmanolane should be "dimethyl. and diethylmanolane derivatives....."
>> Corrected
- In our previous work,[20] should be In our previous work [20],
>> Corrected
- 3% aqueous solution of NaHCO3 should be "Aqueous solution of NaHCO3 (3%, 50 mL)......"
>> Corrected
>> Several other typos were also corrected.
Reviewer 2 Report
The manuscript submitted by Baranov and co-workers describes the reaction of functionalized malonates to produce tetrahydroquinolines THQ.
The authors claimed a new and deep study on the reaction mechanism, which is not the case. The manuscript is a simple extension of the previously reported method of using sulfur instead of nitrogen. Besides, nitrogen was used before fro other authors in very similar transformations.
On the other hand, the manuscript is difficult to follow, and several issues must be addressed.
In summary, I found the chemistry repetitive; no new results or significant finds are reported; the manuscript may be interesting for a limited audience specialized audience and not for the reader of molecules.
Issues:
1. English editing is essential
2. The title and introduction emphasize different species. What is more important for the authors, the boronates or iminium intermediates?
3. The introduction was weirdly written, and the last paragraph is out of context. If the authors want to highlight the significance of THQs, that should be the first paragraph; then, they can show the previous studies related to the current work.
4. In the reaction mechanism, a charge is missing in the Boron atom so that the negative boronate may neutralize the iminium charge. Why do the authors propose a second BF4 anion? Boron and Fluor NMR are needed to clarify that point.
5. A chemical shift of 9.3 ppm for aldiminium proton is only estimated since there is no reference in the reaction media. This should be clarified in the text.
Author Response
Issues:
- English editing is essential
>> Edited. We tried to proofread the text by ourselves and ask for a help of two independent fluent speakers. Many corrections were included in the text.
- The title and introduction emphasize different species. What is more important for the authors, the boronates or iminium intermediates?
>> Formation of both fragments are important. We corrected the corresponding parts of the manuscript to emphasize both. The title was not corrected since it is hard to introduce all information in one sentence.
- The introduction was weirdly written, and the last paragraph is out of context. If the authors want to highlight the significance of THQs, that should be the first paragraph; then, they can show the previous studies related to the current work.
>> Last paragraph of introduction and corresponding Scheme were removed. Some parts of introduction were rewritten.
- In the reaction mechanism, a charge is missing in the Boron atom so that the negative boronate may neutralize the iminium charge. Why do the authors propose a second BF4 anion? Boron and Fluor NMR are needed to clarify that point.
>> The scheme is corrected and negative charge attached. 11B, 19F NMR spectra of the reaction mixture and reference spectra of BF3*OEt2 were added to the supplementary material. Corresponding discussion was included in the text. Heteronuclear NMR data was included in Scheme 2.
- A chemical shift of 9.3 ppm for aldiminium proton is only estimated since there is no reference in the reaction media. This should be clarified in the text.
>> Corrected. We also add 1H NMR spectra of starting material in similar solvent (dichloroethane).
Round 2
Reviewer 2 Report
The authors followed all my former concerns. I still thoink this research is not sufficiently new for a high impact journal as molecules. However, if the Editorial Office give the authors the chance to re-submit their work, i guess there is no reason to reject the paper.
Author Response
Thank you for reviewing our manuscript.
We still believe that our article is important enough for a wide range of readers.
Perhaps the local results are not so impressive - just another method for synthesizing tetrahydroquinoline derivatives. However, the formation of stable charged species from boron trifluoride and arylidene-malonates has not been reported in the literature. It seems to us that such a reaction can be used in other transformations in the future. Moreover, derivatives similar to synthesized 2a* (containing a difluoroboryl bridge, as in BODIPY ) can potentially be used as fluorescent dyes.